# Surface Pre-Reacted Glass-Ionomer Eluate Suppresses Osteoclastogenesis through Downregulation of the MAPK Signaling Pathway

**DOI:** 10.3390/biomedicines12081835

**Published:** 2024-08-12

**Authors:** Janaki Chandra, Shin Nakamura, Satoru Shindo, Elizabeth Leon, Maria Castellon, Maria Rita Pastore, Alireza Heidari, Lukasz Witek, Paulo G. Coelho, Toshiyuki Nakatsuka, Toshihisa Kawai

**Affiliations:** 1Department of Oral Science and Translational Research, College of Dental Medicine, Nova Southeastern University, Fort Lauderdale, FL 33328, USA; jc4409@mynsu.nova.edu (J.C.); sshindo1@nova.edu (S.S.); eleon@mynsu.nova.edu (E.L.); mc4155@mynsu.nova.edu (M.C.); mpastore@nova.edu (M.R.P.); aheidari@nova.edu (A.H.); 2Department of Pathophysiology-Periodontal Science, Faculty of Medicine, Dentistry and Pharmaceutical Sciences, Okayama University, Okayama 700-8525, Japan; 3Biomaterials Division, NYU Dentistry, New York, NY 10010, USA; lw901@nyu.edu; 4Department of Biomedical Engineering, NYU Tandon School of Engineering, Brooklyn, NY 11201, USA; 5Hansjörg Wyss Department of Plastic Surgery, NYU Grossman School of Medicine, New York, NY 10016, USA; 6Department of Biochemistry and Molecular Biology, Miller School of Medicine, University of Miami, Miami, FL 33146, USA; pgc51@med.miami.edu; 7DeWitt Daughtry Family Department of Surgery, Division of Plastic Surgery, Miller School of Medicine, University of Miami, Miami, FL 33146, USA; 8R&D Department, Shofu Inc., Kyoto 605-0983, Japan; t-nakatsuka@shofu.co.jp

**Keywords:** S-PRG, osteoclast, hydroxyapatite, TRAP staining, bioactive filler

## Abstract

Surface pre-reacted glass-ionomer (S-PRG) is a new bioactive filler utilized for the restoration of decayed teeth by its ability to release six bioactive ions that prevent the adhesion of dental plaque to the tooth surface. Since ionic liquids are reported to facilitate transepithelial penetration, we reasoned that S-PRG applied to root caries could impact the osteoclasts (OCs) in the proximal alveolar bone. Therefore, this study aimed to investigate the effect of S-PRG eluate solution on RANKL-induced OC-genesis and mineral dissolution in vitro. Using RAW264.7 cells as OC precursor cells (OPCs), TRAP staining and pit formation assays were conducted to monitor OC-genesis and mineral dissolution, respectively, while OC-genesis-associated gene expression was measured using quantitative real-time PCR (qPCR). Expression of NFATc1, a master regulator of OC differentiation, and the phosphorylation of MAPK signaling molecules were measured using Western blotting. S-PRG eluate dilutions at 1/200 and 1/400 showed no cytotoxicity to RAW264.7 cells but did significantly suppress both OC-genesis and mineral dissolution. The same concentrations of S-PRG eluate downregulated the RANKL-mediated induction of OCSTAMP and CATK mRNAs, as well as the expression of NFATc1 protein and the phosphorylation of ERK, JNK, and p38. These results demonstrate that S-PRG eluate can downregulate RANKL-induced OC-genesis and mineral dissolution, suggesting that its application to root caries might prevent alveolar bone resorption.

## 1. Introduction

The prevalence of periodontal disease is estimated to affect approximately 48% of U.S. adults aged 30 and older [1], and severe periodontal disease is estimated to affect ~11% of the global population [2]. Severe periodontal disease causes the destruction of periodontal tissue, such as alveolar bone and periodontal ligament, which leads to tooth loss [3]. Tooth loss diminishes the quality of life through masticatory disturbances and malnutrition, and periodontitis is associated with systemic diseases [4], such as diabetes [5] and rheumatoid arthritis [6]. Emerging evidence has highlighted the link between inflammation and bacterial infection in periodontal disease and the development or exacerbation of systemic diseases [7,8]. Therefore, it is crucial to not only control infection but also take steps to prevent the destruction of periodontal tissues, such as that seen in alveolar bone resorption. The conventional approach to treating periodontal disease involves the mechanical removal of bacterial biofilm [9], followed by the application of chemotherapeutic agents [10]. Local and systemic administration of antimicrobial agents, such as tetracyclines, has been employed [11,12]. In addition, the efficacy of anti-collagenolytic compounds [10] and resolvins [13,14] has been demonstrated in addressing inflammatory responses in periodontal disease.

Recently, metal ions have demonstrated a bactericidal effect in treating periodontitis [15]. An innovative dental filling material, surface pre-reacted glass ionomer (S-PRG) filler, has been developed by Shofu, Inc. (Kyoto, Kyoto, Japan). S-PRG filler consists of a three-layered fine glass particle, including a layer coated with SiO_2_, a pre-reacted glass-ionomer phase, and fluoroboroaluminosilicate core glass. This composition allows the release of six different ions: fluoride (F^−^), strontium (Sr^2+^), sodium (Na^+^), borate (BO_3_^3−^), aluminum (Al^3+^), and silicate (SiO_3_^2−^) [16,17]. Notably, F^−^, Sr^2+^, and BO_3_^3−^ concentrations are relatively high among the released ions [16], surpassing those from conventional glass-ionomer cement, as previously reported [18]. Recently, a few emerging studies have reported on the functions and efficacy of S-PRG. Specifically, it has been shown to strengthen tooth structure and inhibit tooth demineralization [19], exhibit buffering capacity against oral acidity [17], and reduce oral plaque adhesion to tooth surfaces [20]. In commercial production since 2000, products containing S-PRG have been applied clinically as adhesive systems, temporary cements, composite resins, fissure sealants, and polishing pastes. The use of S-PRG filler in tooth restoration restores occlusal function and protects against hypersensitivity. Moreover, several studies have reported on the bactericidal effect of S-PRG fillers and eluates [21,22,23]. S-PRG eluate has been found to inhibit the growth of *Porphyromonas gingivalis* (*Pg*), a keystone pathogen in periodontal disease, by suppressing its protease activity [20,22]. Ultrasonic irrigation with S-PRG filler dispersion decreased the ratio of periodontal pathogens in the bacterial flora of periodontal tissue [24]. Additionally, S-PRG eluate reportedly inhibits alveolar bone loss and maintains bone density by reducing the destruction of the collagen bundle in the periodontal ligament and lowering the infiltration of inflammatory cells in the ligature-induced periodontitis mouse model [25]. Furthermore, S-PRG nanofiller coating reportedly exhibits antimicrobial activity, improving periodontitis in a canine model [26]. These findings suggest that S-PRG eluate has a prophylactic effect against tissue destruction in periodontal disease by the reduction of inflammatory response. S-PRG has also been found to induce osteoblast differentiation [27] and tertiary dentin formation from dental pulp stem cells [28], indicating its effectiveness in pulp capping [29] and bone regeneration [30].

As previously mentioned, S-PRG fillers are used as filling materials, fissure sealants, and coating materials to protect and restore tooth decay. The ability of S-PRG fillers to modulate the pH of the surrounding medium toward neutral or weak alkaline values [17] and the release of F^−^ and Sr^2+^ ions all contribute to tooth strength, making S-PRG effective in caries management. As the aging of society advances, the prevalence of root caries has been on the rise. Reports indicate that 10–20% of root caries lesions occur below the gumline [31] and that their onset is associated with periodontal disease [32]. Serving as a filling material for root caries, S-PRG fillers release multiple ions that impact the proximal alveolar bone. Maintaining bone metabolism is crucial in preserving a homeostatic balance between osteoblasts (OBs) and osteoclasts (OCs), which is, in turn, vital for the coupling between bone resorption and bone formation. Disruption of this balance can lead to pathogenic bone resorption [33]. OCs, multinucleated cells derived from hematopoietic stem cells through monocyte/macrophage lineage precursors, play a key role in bone resorption. During the process of osteoclastogenesis (OC-genesis), cells progress through several stages to become mature OCs [34,35].

A hallmark of periodontitis is bone resorption from local differentiation and activation of OCs promoted by infection and inflammation. Consequently, the regulation of pathogenic OCs activation plays an important role in suppressing the progression of periodontal disease. Although topically applied S-PRG to a pulp chamber has been reported to indirectly decrease the number of OCs in the periapical lesions induced in a rat model of pulp-exposure-induced apical periodontitis [36], the direct effect of S-PRG on OC-genesis remains unclear. Therefore, in this study, we aimed to elucidate the effect of multiple ions released from S-PRG filler on OC-genesis. To accomplish this, we prepared an S-PRG eluate and investigated its potential regulation of OC-genesis in vitro. Our results collectively supported the benefits of S-PRG in preventing bone resorption in the alveolar bone adjacent to root surface dental caries filled with tooth restorative material.

## 2. Materials and Methods

### 2.1. Preparation of S-PRG Eluate

S-PRG eluate was provided by Shofu, Inc. (Kyoto, Japan) and prepared as previously described [17]. Briefly, S-PRG filler (1 µm) was mixed with distilled water in a 1:1 ratio and gently stirred at room temperature for 24 h. The mixture underwent filtration and centrifugation to eliminate any insoluble materials.

### 2.2. Chemical Reagents

To examine the effect of individual ion in S-PRG eluate on OC-genesis, the following chemicals were used to create respective ion solution; B(OH)_3_, NaF, AlCl_3_, SrCl_2_ and Na_2_SiO_3_ (Sigma Aldrich, St. Louis, MO, USA). PD98059 (ERK inhibitor), SB203580 (p38 inhibitor), or SP600125 (JNK inhibitor) were purchased from Cell Signaling Technology (Danvers, MA, USA).

### 2.3. Cell Viability Assay

To evaluate the possible cytotoxicity of S-PRG eluate, the viability of a murine macrophage-like cell line, RAW264.7 OC precursor cells (OPCs) (ATCC, Manassas, VA, USA) was measured using WST-8. Cells were cultured in Dulbecco’s Modified Eagle’s Medium (DMEM, Thermo Fisher Scientific, Waltham, MA, USA) containing 10% fetal bovine serum (FBS) at 37 °C, 5% CO2, and 95% humidity and subsequently seeded in DMEM containing 10% FBS in wells of a 96-well plate (Corning, Inc., Corning, NY, USA) at a density of 3.0 × 10^3^ cells/well. After 24 h of incubation, S-PRG eluate was diluted at 1/10, 1/50, 1/100, 1/200, 1/400, 1/800, 1/1000, and 1/1200 with Phosphate-buffered saline (PBS, pH 7.4, Thermo Fisher Scientific) added to each well. WST-8 assay was conducted using the Cell Counting Kit-8 (Abcam, Cambridge, MA, USA) at 24 h after S-PRG eluate addition to assess cell cytotoxicity.

### 2.4. TRAP Staining

RAW264.7 cells were cultured using a 96-well plate at a density of 3.0 × 10^3^ cells/well. After 24 h of incubation, the dilutions of S-PRG eluate that showed no toxicity were added with 10 ng/mL of receptor activator of nuclear factor κβ Ligand (RANKL, BioLegend, San Diego, CA, USA). After 5 days, tartrate-resistant acid phosphatase (TRAP) staining (Sigma-Aldrich, St. Louis, MO, USA) was performed. Similarly, after 24 h of pre-incubation in a 96-well plate, respective ions (Sr, Al, F, B, or Si ions) in S-PRG eluate were individually applied to the RAW264.7 cells incubated with or without RANKL (10 ng/mL). The concentration of each ion added to the RAW264.7 cell culture was a 200-fold dilution of that reported by Ito et al., who reported the concentrations of 6 ions in S-PRG eluate [16] (Table 1). TRAP-positive cells with more than 3 nuclei were considered mature OCs and were counted using a light microscope (EVOS™ XL Core Imaging System, Thermo Fisher Scientific). Images from the samples were obtained using the EVOS™ XL Core Imaging System (Thermo Fisher Scientific).

### 2.5. Pit-Formation Assay

RAW264.7 cells were plated in hydroxyapatite (HA)-coated wells of a 96-well plate prepared for pit-formation assay at a density of 3.0 × 10^3^ cells/well. After 24 h, cells were stimulated with S-PRG eluate and 10 ng/mL RANKL for 7 days. Cells were removed with 10% bleach and observed under light microscopy. Images from the samples were obtained using the EVOS™ XL Core Imaging System (Thermo Fisher Scientific). Pit area was measured using public domain software (ImageJ 1.54h 15 December 2023, National Institutes of Health, Bethesda, MD, USA).

### 2.6. Quantitative Real-Time PCR

RAW264.7 cells were seeded in wells of a 12-well plate at a density of 3.5 × 10^4^ cells/well. After 24 h, S-PRG eluate was added with 10 ng/mL RANKL. Cells were lysed after 24 h, and mRNA was isolated using the PureLink™ RNA Mini Kit (Thermo Fisher Scientific). mRNA concentration was determined by measuring optical density at 260 nm. To ensure the purity of isolated mRNA, A260/280 was confirmed to be in the range of 1.8–2.0 with Nanodrop (Thermo Fisher Scientific). Using 10 ng of mRNA, cDNA was synthesized using the VERSO cDNA kit (Thermo Fisher Scientific). Polymerase Chain Reaction (PCR) was carried out at 95 °C for 20 s, then 40 cycles at 95 °C for 1 s, and 4 cycles at 60 °C for 20 s. Glyceraldehyde-3-phosphate dehydrogenase (GAPDH) was used as the internal control gene to normalize the mRNA level. The gene expression of ocstamp (Mm00512445_m1), dcstamp (Mm04209236_m1), nfatc1 (Mm01265944_m1), acp5 (Mm00475698_m1), and cathepsinK (Mm00484039_m1) was examined.

### 2.7. Immunoblotting for NFATc1

RAW264.7 cells were seeded using in wells of a 12-well plate at a density of 3.5 × 10^4^ cells/well. After 24 h, S-PRG eluate was added with 10 ng/mL RANKL. Cells were lysed after 24 h using M-PER™ Mammalian Protein Extraction Reagent (Thermo Fisher Scientific) supplemented with Halt Protease Inhibitor Cocktails diluted 100× (Thermo Fisher Scientific). The concentration of protein was assessed with the Pierce™ BCA assay kit (Thermo Fisher Scientific). Proteins were mixed with Sodium dodecyl sulfate (SDS) solution and reducing buffer and then heated at 70 °C for 10 min. SDS-polyacrylamide gel electrophoresis (SDS-PAGE) was performed, and the proteins were transferred to polyvinylidene difluoride (PVDF) membrane using the iBlot 3Western blot transfer system (Thermo Fisher Scientific). After blocking the membrane with Tris-Buffered Saline with 0.1% Tween20 Detergent (TBST) containing 2% skim milk, we applied mouse anti-NFATc1, which is a master transcription regulator of OC differentiation (sc-7296, Santa Cruz Biotechnology, Santa Cruz, CA, USA), diluted at 1:1000 or rabbit anti-GAPDH (#5174, Cell Signaling Technology, Beverly, MA, USA) diluted at 1:1000 with TBST containing 2% skim milk. After 16 h of incubation at 4 °C, the membrane was washed with TBST, and horseradish peroxidase (HRP)-labeled anti-rabbit or anti-mouse antibody (Cell Signaling Technology) as a secondary antibody was used for 1 h. The membrane was washed with TBST and peroxidase substrate for enhanced chemiluminescence (Thermo Fisher Scientific) detection. The image was obtained using the Azure C400 (Azure Biosystems, Inc., Dublin, CA, USA). Relative density was calculated by comparing NFATc1 and GAPDH using ImageJ.

### 2.8. Immunoblotting for MAPK and NF-κB Signaling

RAW264.7 cells were seeded in a 12-well plate at a density of 3.5 × 10^4^ cells/well. After 24 h of pre-incubation, S-PRG eluate was added with RANKL (10 ng/mL). After 30 min, cells were lysed, and Western blotting was carried out using the same method described above. All primary antibodies, including pERK (#9101), ERK (#4695), pJNK (#9251), JNK (#9252), pp38 (#4511), p38 (#8690), IκBα (#4814), and GAPDH (#5174), were obtained from Cell Signaling Technology. HRP-labeled anti-rabbit secondary antibody (Cell Signaling Technology) was applied, and the membrane was detected using the device. Relative density was calculated by comparing phosphorylated protein and total protein using ImageJ.

### 2.9. Assay to Test the Effect of MAPK Inhibitors on OC-Genesis

RAW264.7 cells were cultured in wells of a 96-well plate at a density of 3.0 × 10^3^ cells/well. After 24 h of pre-incubation, cells were stimulated with PD98059 (ERK inhibitor), SB203580 (p38 inhibitor), and SP600125 (JNK inhibitor) with or without RANKL (10 ng/mL). TRAP staining was carried out on Day-5, while resorption pit formation assay was conducted on Day-7. Moreover, S-PRG eluate along with MAPK inhibitors was applied to RAW264.7 cells, and the number of TRAP-positive cells was counted.

### 2.10. Release Assay

To evaluate the effect of S-PRG released from HA on OC-genesis, a release assay was conducted. S-PRG eluate was coated on an HA plate for 24 h and then replaced with DMEM. The supernatant was collected at different time points (0.25, 0.5, 1, 3, 6, and 12 h) and applied to an OC-genesis assay plate (Figure 1). TRAP staining was performed after 5 days, and TRAP-positive cells were counted.

### 2.11. Statistical Analysis

All quantitative data are represented as the means ± standard deviation (SD). One-way ANOVA with Tukey–Kramer multiple comparison test was carried out. For each statistical process, a test was conducted using EZR (Easy R Ver 1.68) software (EZR, Saitama Medical Center, Jichi Medical University, Saitama, Japan), and a *p*-value of <0.05 was considered statistically significant [37].

## 3. Results

### 3.1. Cytotoxicity of S-PRG Eluate

To determine the optimal concentration of S-PRG eluate, a WST-8 assay was performed. S-PRG eluate showed no cytotoxicity by dilution at greater than 1/200 (Figure 2). Consequently, we used S-PRG eluate diluted at more than 1/200 in subsequent experiments.

### 3.2. Effect of S-PRG Eluate on OC-Genesis

TRAP staining and pit-formation assay were conducted to assess the effect of S-PRG eluate on OC-genesis. The results indicated that S-PRG eluate, when diluted at 1/200, 1/400, 1/600, and 1/1000, significantly decreased the number of TRAP-positive cells with more than 3 nuclei compared to the control (Figure 3A,B). Additionally, S-PRG eluate decreased the pit area generated by mature OCs compared to the control (Figure 3C,D).

### 3.3. Effect of S-PRG Eluate on OC-Genesis-Related Genes and Protein Expression

The expression of genes associated with OC-genesis, including ocstamp, dcstamp, acp5, cathepsinK, and nfatc1, was examined using quantitative RT-PCR. S-PRG eluate, diluted at 1/200, significantly suppressed the expression of ocstamp, dcstamp, cathepsinK, and nfatc1 compared to the control. The expression of ocstamp, acp5, cathepsinK, and nfatc1 upon addition of S-PRG eluate diluted at 1/400 was significantly suppressed compared with control (Figure 4A). Moreover, the expression of NFATc1, which is a master transcription regulator of OC differentiation, was evaluated using Western blotting. Within 24 h of adding S-PRG eluate diluted at 1/200 and 1/400 with RANKL, the expression of NFATc1 significantly decreased compared to the control (Figure 4B,C).

### 3.4. Evaluation of RANKL-Induced MAPK and NF-κB Activation Regulated by S-PRG Eluate

To investigate the effect of S-PRG eluate on signal transduction in OC differentiation, the phosphorylation of MAPK proteins and NF-κB protein, along with IκBα, a member of a family of cellular proteins that inhibit NF-κB, was examined with Western blotting. S-PRG eluate diluted at 1/200 and 1/400 significantly decreased the phosphorylation of ERK, JNK, and p38 within 30 min of adding RANKL compared to the control (Figure 5A,B). Additionally, despite RANKL stimulation leading to a decrease in the expression of total IκBα, S-PRG eluate diluted at 1/200 significantly restored its expression (Figure 5C,D).

### 3.5. Effect of MAPK Inhibitors on OC-Genesis

To assess the possible effects of MAPK inhibitors on RANKL-induced OC-genesis, PD98059 (ERK inhibitor), SB203580 (p38 inhibitor), or SP600125 (JNK inhibitor) was applied to Raw264.7 cells stimulated with RANKL (10 ng/mL) and allowed to culture for appropriate periods. Then, TRAP staining and pit formation assay were performed. All inhibitors used significantly decreased the number of TRAP-positive cells and pit area (Figure 6A–D).

Additionally, we assessed the combined effect of S-PRG eluate and MAPK inhibitors on TRAP-positive cells. S-PRG eluate at 1/200 and 1/400 added to PD98059, SB203580, and SP600125 decreased the number of TRAP-positive cells compared to the negative control (NC). S-PRG eluate added to PD98059 or SP600125 decreased the number of TRAP-positive cells decreased compared to PD98059 or SP600125 alone (Figure 7).

### 3.6. Regulation of OC-Genesis by S-PRG Eluate Released from HA

The effect of S-PRG eluate released from HA on OC-genesis was investigated with TRAP staining. Supernatant from the HA-coated plate treated with S-PRG eluate revealed a significant decrease in the number of TRAP-positive cells from 0.25 to 12 h compared with control (Figure 8).

### 3.7. Effect of Each Ion in S-PRG Eluate on OC-Genesis

The possible effect of each ion contained in S-PRG eluate was examined using TRAP staining and pit formation assay. Sr^2+^, Al^3+^, F^−^, and B^3+^ significantly suppressed the number of TRAP-positive cells (Figure 9A,B). In the pit formation assay, Sr^2+^, F^−^, and B^3+^ significantly ameliorated the pit area compared to the control group (Figure 9C,D).

## 4. Discussion

Periodontal disease, an oral inflammatory disease, is characterized by the destruction of periodontal tissue by the interaction between bacterial infection and host immune response. Severe periodontal disease results in bone resorption caused by pathologically activated OCs [38]. In this work, we focused on S-PRG, which has been utilized clinically through its incorporation into adhesive systems and orthodontic resins. S-PRG can release six types of ions (BO_3_^3−^, Na^+^, Al^3+^, SiO_3_^2−^, Sr^2+^, and F^−^) able to change repeatedly, suggesting that S-PRG has long-term effects [18,39]. According to Imazato et al., products containing S-PRG filler exerted continuous release of multiple ions for 440 days [18]. Furthermore, owing to the different functions of these ions, reports have shown that S-PRG eluate has antibacterial activity and can inhibit oral biofilm formation [22], leading to the suppression of inflammatory response in periodontal tissue [25]. In addition, S-PRG eluate-induced osteoblast (OB) differentiation from mesenchymal stem cells [40]. Accordingly, these studies suggest that released ions from S-PRG materials could diffuse throughout surrounding tissue with resultant efficacy against the development and progression of periodontal disease. However, the effect of S-PRG eluate on OC-genesis remains elusive. Based on the results of previous studies, we reasoned that the release of multiple ions from S-PRG might have an effect on OCs in surrounding periodontal tissues and, consequently, the suppression of OC-genesis.

The work has also confirmed that the dilution of S-PRG eluate over 200-fold resulted in cell growth comparable to that of the positive control with no cytotoxicity to OCs. Therefore, we used S-PRG eluate diluted at 1/200 or more, and a 200- to 1000-fold dilution of aqueous S-PRG eluate significantly decreased the number of TRAP-positive cells with more than 3 nuclei. Furthermore, a 200- and 400-fold dilution of S-PRG eluate decreased pit area and the expression of OC-genesis-related genes induced by RANKL. Furthermore, S-PRG eluate suppressed the expression of NFATc1, which is known as a master gene regulating OC differentiation [41]. These results show that S-PRG eluate could effectively suppress OC differentiation in vitro.

The activation of MAPKs and NFATc1 by RANKL reportedly contributes to OC precursor survival, proliferation, and OC differentiation [41,42]. MAPKs play a pivotal role in the transduction of cell signals that regulate diverse cellular activities, including gene expression, cell cycle progression, metabolism, motility, survival, apoptosis, and differentiation [43,44]. OC differentiation can be influenced by a more persistent biphasic activation of MAPKs by RANKL, whereby RANKL-mediated activation of p38 plays a prominent role in promoting OC-genesis [45]. In addition, we observed the suppressive effect of MAPK inhibitors on OC-genesis, suggesting that ERK, JNK, and p38 are engaged in the RANKL-mediated OC-genesis. However, in our study, S-PRG eluate significantly decreased the phosphorylation of ERK, JNK, and p38. Moreover, when we combined S-PRG eluate with PD98059 and SP600125, OC-genesis was significantly suppressed compared to inhibitors alone. Therefore, it could be concluded that S-PRG eluate exerts an additive effect in combination with MAPK inhibitors, suggesting that S-PRG eluate can specifically inhibit ERK and JNK. However, RAW264.7 cells used in this study are OCPs capable of proliferating without macrophage colony-stimulating factor (M-CSF) treatment and differentiating into mature OCs in response to RANKL treatment alone. M-CSF can also induce the MAPK activation of OCPs [45]. Therefore, further investigation using primary cells, such as bone marrow-derived monocytes, is needed to confirm the anti-OC-genesis effect of S-PRG in the physiological context. On the other hand, NF-κB has been reported to induce the expression of NFATc1, which promotes the differentiation of OCs differentiation in the early stage of OC-genesis [46]. Upon RANKL stimulation, the phosphorylation of IKK is induced, leading to ubiquitin-dependent degradation of IκBα, a member of a family of cellular proteins that function to inhibit the NF-κB transcription factor [47]. Our results did, indeed, reveal a decrease in total IκBα following RANKL stimulation; however, we were encouraged by its restoration by S-PRG eluate. Therefore, despite the caveat noted above, these results suggest that S-PRG eluate can regulate MAPK and NF-κB activation, contributing to the suppression of OC-genesis.

Released ions from S-PRG were reportedly absorbed in hydroxyapatite and then discharged [39]. Taking advantage of this phenomenon, our experiments examined the effect of S-PRG eluate released after absorption in hydroxyapatite on OC-genesis. The supernatant collected at different time points from S-PRG-treated HA showed a decrease in the number of TRAP-positive cells. These results show that multiple ions contained in S-PRG eluate are first uptaken in HA and then released to suppress OC-genesis. Since teeth and bone are composed of HA, these findings, in turn, suggest that ions released from S-PRG can be incorporated into both teeth and alveolar bone and that such ions released from bone can suppress local OC-genesis.

Although S-PRG contains six ions, their specific functions in the regulation of OC differentiation remain elusive. It has been reported that S-PRG filler contained Na, 498 ppm; Si, 52 ppm; Sr, 319 ppm; B, 1989 ppm; Al, 86 ppm; and F, 201 ppm (Table 1, 16), whereas individual ion content in S-PRG eluate was Na, 3350.6 ppm; Si, 7.3 ppm, Sr, 792.8 ppm; B, 1456 ppm; Al, 12 ppm; and F, 54.3 ppm [48], indicating that Na, Sr, B and F ions are also profoundly released from S-PRG filler in the physiological context too. Given the large amount of Na (9300 ppm) present in the medium to maintain physiological osmolality, we sought to address the effects of Sr, B, and F on dysregulated bone remodeling in periodontitis where OCs play key role in bone resorption. Some studies have reported that Sr can regulate OC-genesis. For instance, strontium ranelate can induce OC-apoptosis through the calcium-sensing receptor [49] and thus be used to treat postmenopausal osteoporosis [50]. Furthermore, while Sr-substituted bioactive glasses reportedly inhibited OC-genesis through the activation of p38 [51], Sr also inhibited titanium particle-induced OC activation via suppression of the NF-κB pathway, including IκBα [51,52]. Our results showed that S-PRG eluate suppressed MAPK signaling, including the phosphorylation of p38, thereby counteracting the decrease in total IκBα induced by RANKL stimulation, suggesting that the impact of S-PRG eluate might be attributable to Sr. However, the other ions in S-PRG eluate, i.e., BO_3_^3−^ SiO_3_^2−^ Al^3+^ and F^−^, also appear to possess an anti-bone resorptive effect. BO_3_^3−^ has been reported to inhibit alveolar bone loss in rats through diminished bone resorption [53]. In addition, Boric acid inhibits OC-genesis via the suppression of PERK-eIF2α signaling and ameliorates LPS-induced bone loss in vivo [54], whereas boric acid gel shows efficacy in reducing pocket depth and regaining the clinical attachment in periodontitis [55]. SiO_3_^2−^ inhibits RANKL-induced OC-genesis in RAW264.7 cells [56] and improves ovariectomy-induced bone loss [57]. Aluminum contained in mineral trioxide aggregate inhibits OC-genesis [58]. Fluoride (0.5 mg/L) significantly decreased the activity of osteoclast bone resorption derived from mouse bone marrow [59]. These lines of evidence indicate that all ions in S-PRG, not just Na^+^, may exhibit an anti-bone resorptive effect by acting on osteoclasts. Therefore, we investigated the individual effect of multiple ions incorporated in S-PRG on OC-genesis, and our results showed that 4 ions, except SiO_3_^2−^, clearly inhibited OC-genesis. These findings strongly suggest that the multiple ion content found in S-PRG eluate might work in concert to suppress OC-genesis, but further confirmatory studies are required to fully understand the regulatory mechanism underlying such inhibitory effects.

It is well known that S-PRG can be used to treat and prevent periodontal disease through antimicrobial and anti-inflammatory effects in a canine model [24,25,26]. Our study has demonstrated the direct downregulatory effect of S-PRG eluate on OC-genesis in addition to the previously reported anti-demineralization activity, antimicrobial activity, and anti-inflammatory effect, suggesting its possible efficacy against periodontal disease (Figure 10). S-PRG eluate specifically inhibited OC-genesis by downregulating NFATc1 via MAPK and NF-κB signaling. Additionally, our results suggested that ions released from S-PRG filler could be trapped by bone hydroxyapatite, thereby becoming a secondary reservoir for the release of those host-beneficial ions in inhibiting OC-genesis. These findings suggest that the inhibition of OC-genesis not only arises from the various ions released from S-PRG filler mixed in the root cavity filling material but also results from the release of these ions by bone hydroxyapatite as a secondary reservoir. Such a mutual rechargeability between S-PRG and bone hydroxyapatite is anticipated to facilitate the potential efficacy of S-PRG in preventing and treating periodontitis. Taken together, S-PRG eluate could be applied as a potential therapeutic and prophylactic approach for dental diseases, including root caries and periodontitis, that cause alveolar bone resorption in the proximal subgingival root surface.

## 5. Conclusions

Our study established that S-PRG eluate has unique properties against OCs in addition to its remineralization, antimicrobial, and anti-inflammatory activities, as mentioned in previous studies. S-PRG eluate directly suppressed OC-genesis throughout its downregulation of the MAPK/NFATc1 axis, suggesting that S-PRG released from tooth cavity fillings in adjacent periodontal tissue could mitigate the proliferation of OCs in proximal alveolar bone through the release of multiple ions with both therapeutic and prophylactic efficacy on periodontitis.

## Figures and Tables

**Figure 1 biomedicines-12-01835-f001:**
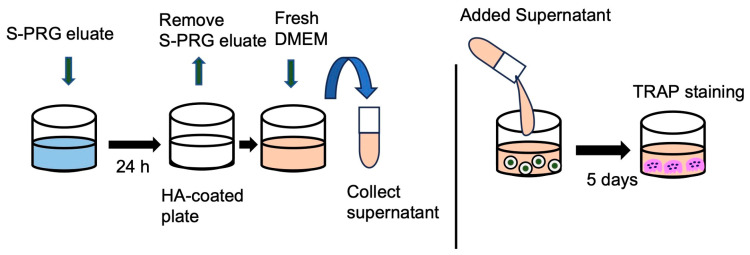
Timeline for release assay. S-PRG eluate was applied to HA-coated plate for 24 h. Fresh media were added to the wells, and the supernatant was collected at different time points. Collected supernatant was applied to OC-genesis assay.

**Figure 2 biomedicines-12-01835-f002:**
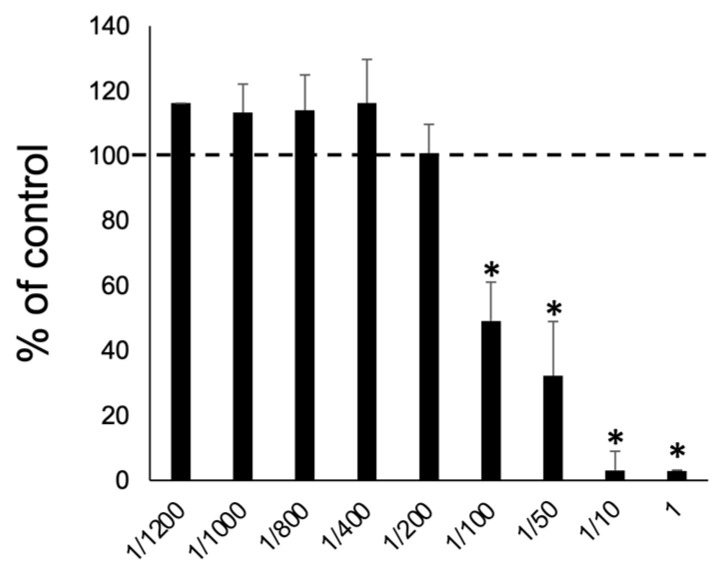
Cytotoxicity of S-PRG eluate. S-PRG eluate had no cytotoxic effect on RAW264.7 cells after diluting to more than 1/200. The horizontal dashed line with a value of 100 indicates the survival rate of RAW264.7 cells cultured in the absence of S-PRG eluate. Results were presented as the means ± SD. *: *p* < 0.05, vs. control without S-PRG eluate.

**Figure 3 biomedicines-12-01835-f003:**
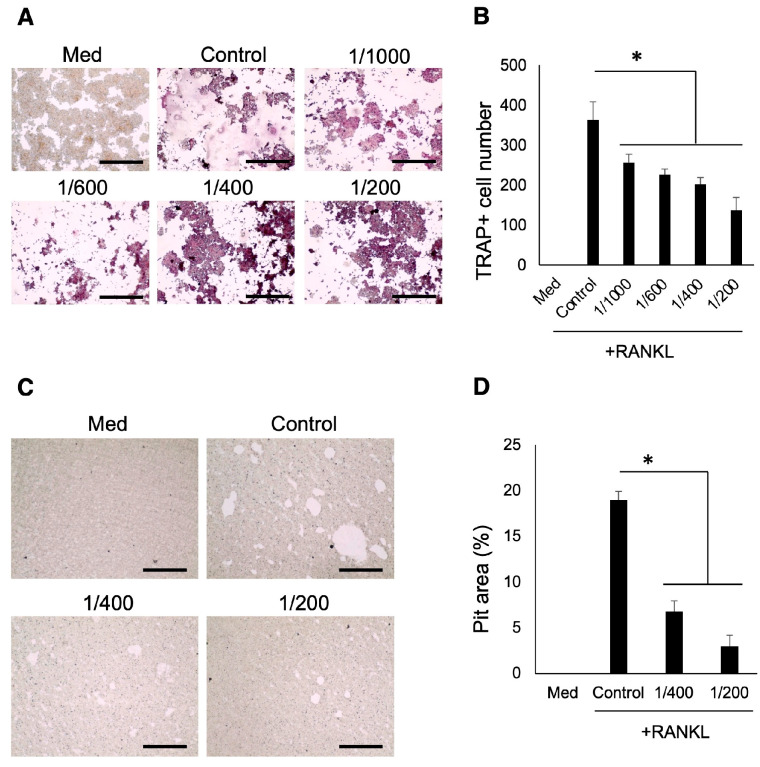
Evaluation of OC-genesis. The number of TRAP-positive cells by adding S-PRG eluate to osteoclast precursors (**A**,**B**). Quantification of pit area when adding S-PRG eluate. S-PRG eluate significantly decreased pit area created by osteoclasts compared to control (**C**,**D**). Scale bars indicate 100 µm. Results were presented as the means ± SD. *: *p* < 0.05, vs. control.

**Figure 4 biomedicines-12-01835-f004:**
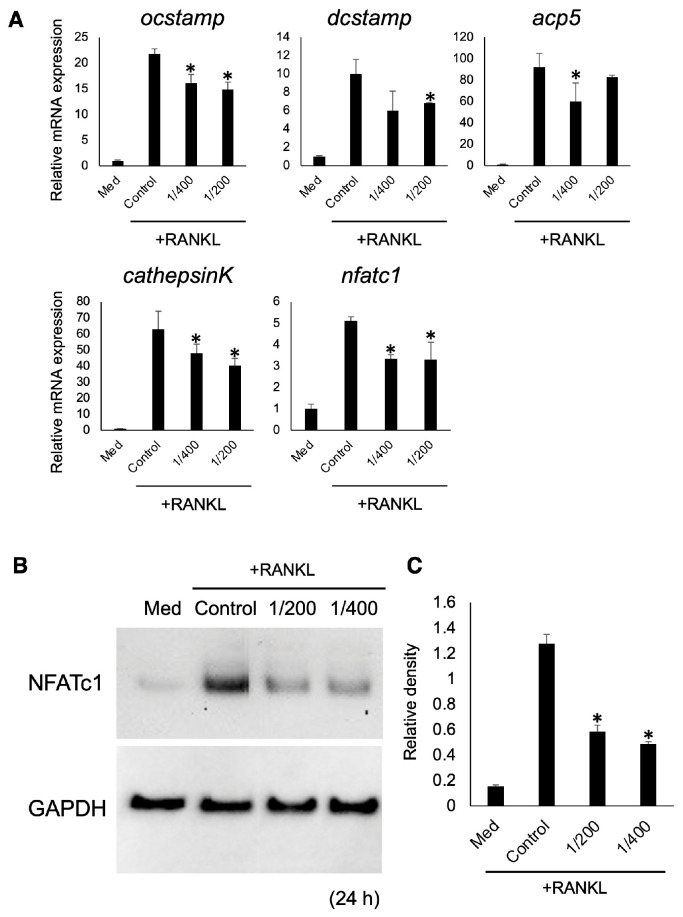
Quantification of mRNA expression related to OC-genesis (**A**) and NFATc1 induction (**B**,**C**) when adding S-PRG eluate. Results were presented as the means ± SD. *: *p* < 0.05, vs. control.

**Figure 5 biomedicines-12-01835-f005:**
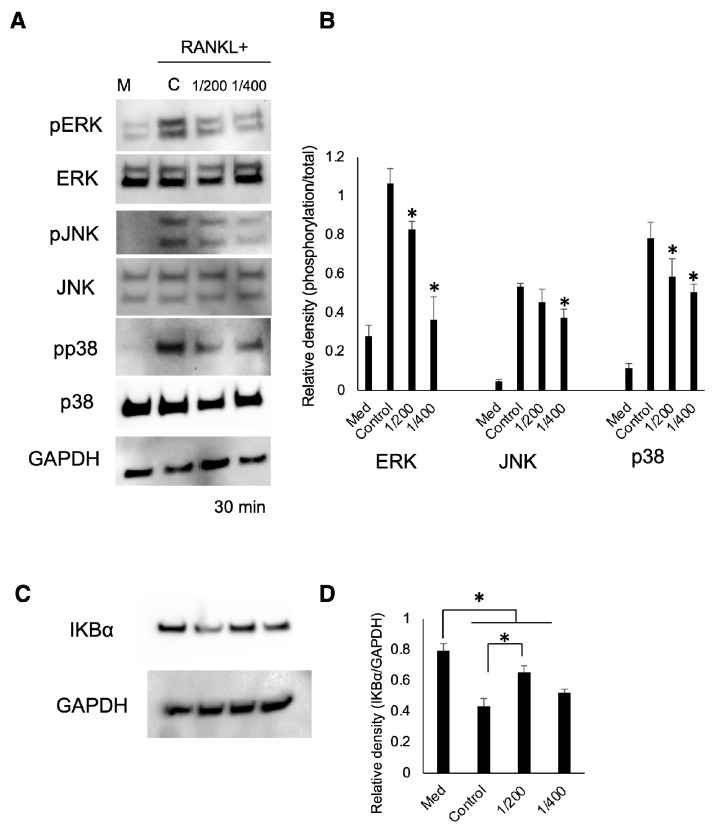
Phosphorylation of MAPK proteins (**A**,**B**) and the expression of total IκBα (**C**,**D**) by adding RANKL and S-PRG eluate. Results were presented as the means ± SD. *: *p* < 0.05, vs. control.

**Figure 6 biomedicines-12-01835-f006:**
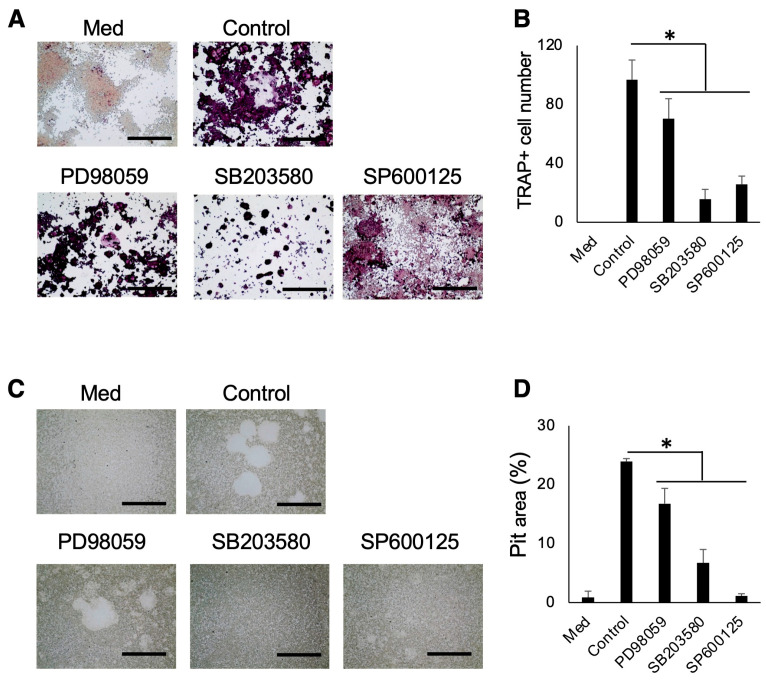
MAPK inhibitors suppressed the OC-genesis. The inhibitors against ERK, p38, and JNK significantly decreased the number of TRAP-positive cells (**A**,**B**) and pit area (**C**,**D**). Scale bars indicate 100 µm. Results were presented as the means ± SD. *: *p* < 0.05, vs. control.

**Figure 7 biomedicines-12-01835-f007:**
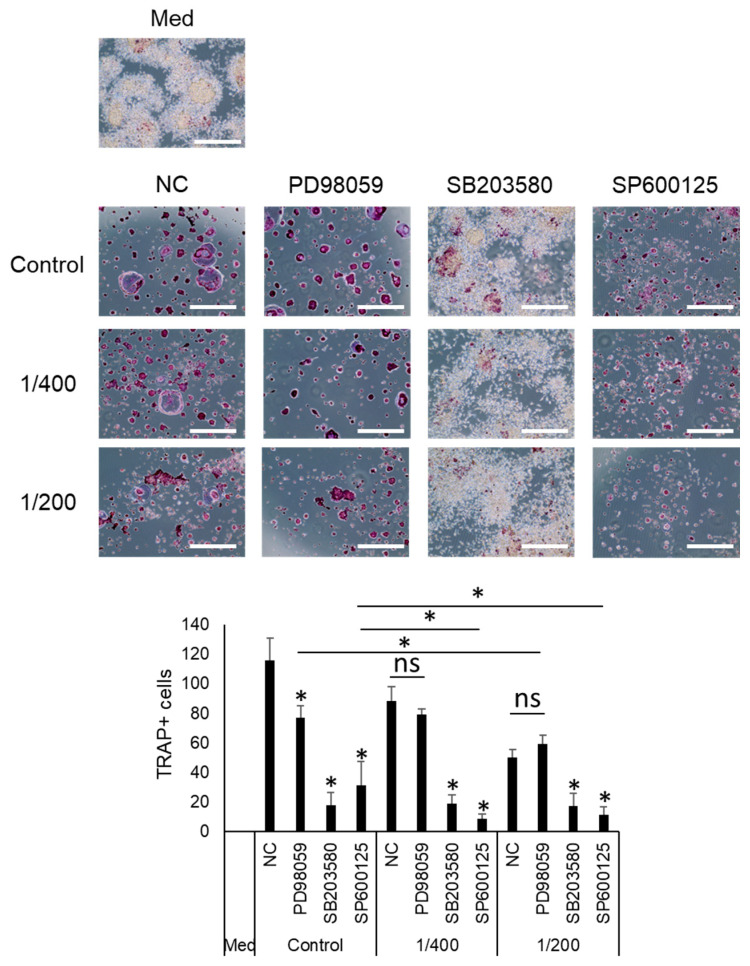
S-PRG eluate, in combination with MAPK inhibitors, suppressed the OC-genesis. The addition of S-PRG elute to inhibitors against ERK, p38, and JNK adding to S-PRG eluate significantly decreased the number of TRAP-positive cells. Results were presented as the means ± SD. Scale bars indicate 100 µm. *: *p* < 0.05, vs. control.

**Figure 8 biomedicines-12-01835-f008:**
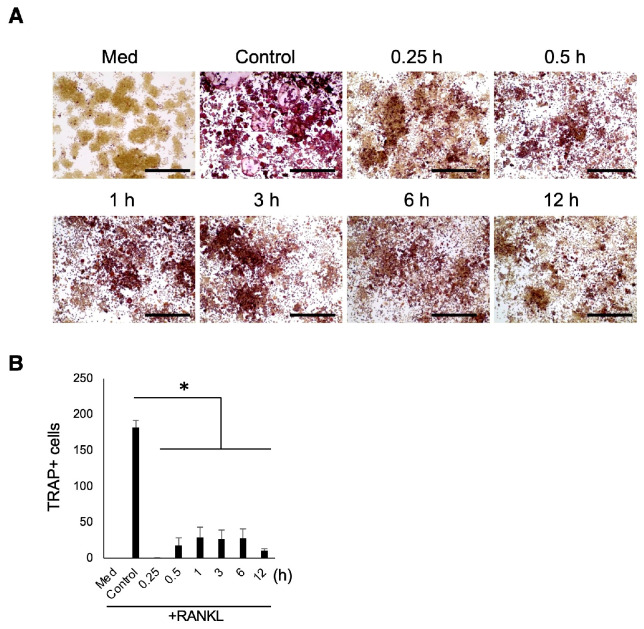
Evaluation of the number of TRAP-positive cells (**A**,**B**) when adding S-PRG eluate released from HA. Scale bars indicate 100 µm. Results were presented as the means ± SD. *: *p* < 0.05, vs. control.

**Figure 9 biomedicines-12-01835-f009:**
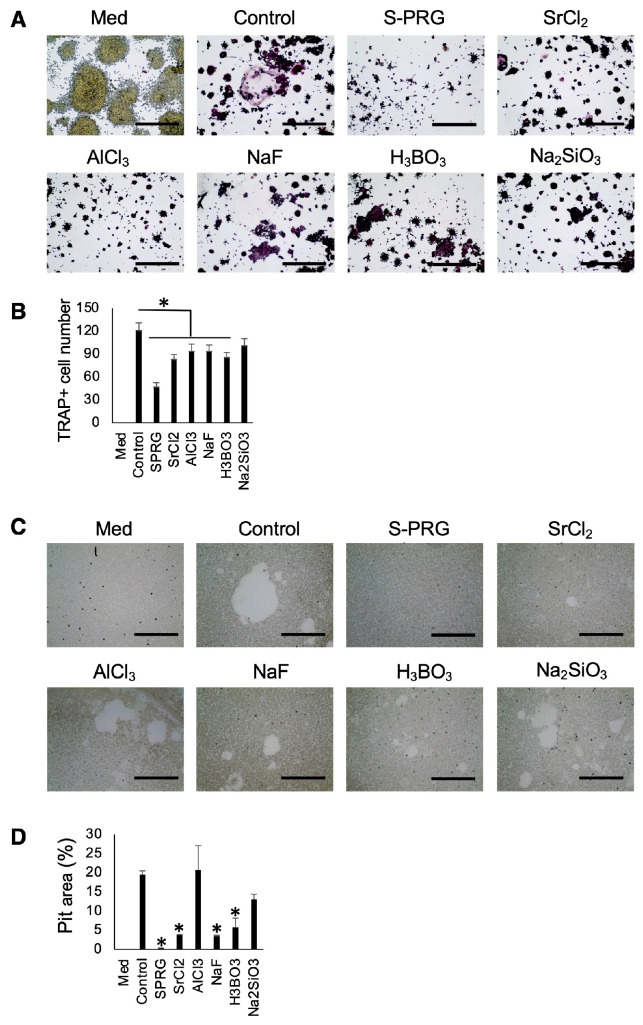
Each ion present in S-PRG eluate inhibited RANKL-induced OC-genesis. Sr^2+^, Al^3+^, F^−^, and B^3+^ significantly decreased the number of TRAP-positive cells induced in Raw264.7 cells stimulated with RANKL (**A**,**B**). Pit area created by Raw264.7 cells incubated with Sr^2+^, F^−^, and B^3+^ in the presence of RANKL was smaller than that in control (**C**,**D**). Scale bars indicate 100 µm. Results were presented as the means ± SD. *: *p* < 0.05, vs. control.

**Figure 10 biomedicines-12-01835-f010:**
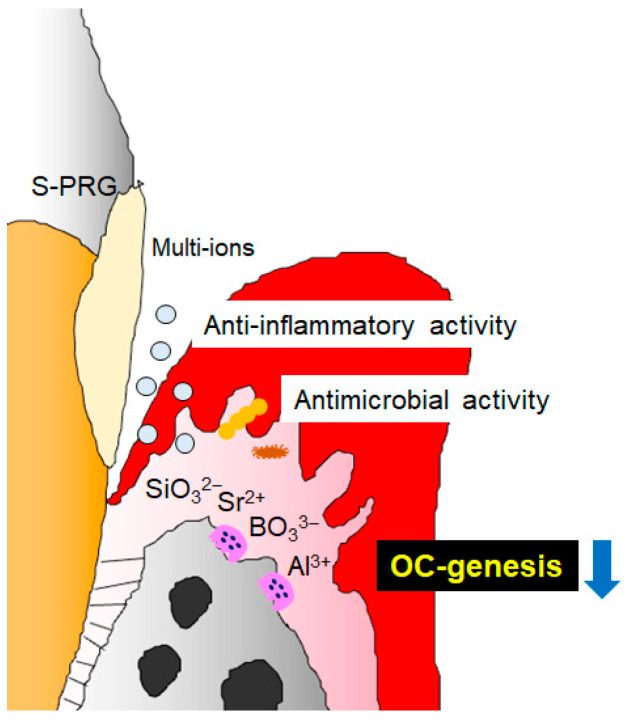
Potential effect of S-PRG eluate on periodontal disease. S-PRG eluate downregulates OC-genesis, as well as anti-inflammatory activity and antimicrobial activity.

**Table 1 biomedicines-12-01835-t001:** The concentration of 6 ions detected in S-PRG eluate. To determine the effect of respective ions in S-PRG eluate upon RANKL-induced OC-genesis, the concentration of 200-fold dilution of that reported by Ito et al. [16] was applied to the RAW264.7 cells incubated with or without RANKL. * Since the Na^+^ ion is necessary to maintain the physiological osmolality, Na^+^ concentration at 9300 ppm was retained in all cultures.

Ion	S-PRG Eluate (ppm; Ito et al., 2011)	Current Study (ppm)
Na	498	9300 *
Si	52	0.26
Sr	319	1.59
B	1989	9.94
Al	86	0.43
F	201	1.01

## Data Availability

All data based on the results are available as part of the article.

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
