# Peer review of "Surface Pre-Reacted Glass-Ionomer Eluate Suppresses Osteoclastogenesis through Downregulation of the MAPK Signaling Pathway"

_biomedicines, 2024, doi:10.3390/biomedicines12081835_

Round 1
Reviewer 1 Report
Comments and Suggestions for Authors
Dear authors,
I read the manuscript: S-PRG Eluate Suppresses Osteoclastogenesis through 2 Downregulation of the MAPK Signaling Pathway , and it can be considered interesting.
The manuscript presents some leaks in its structure, that I advice the authors to rephrase.
Title
S-PRG Eluate Suppresses Osteoclastogenesis through Downregulation of the MAPK Signaling Pathway
The title needs to be changed. What is S-PRG?
Materials and Methods
Line 113: “…RAW264.7 OC precursor cells…”
What is this cell line?
Line 129: “cells…were counted under light microscopy…”
What is the microscopy used?
Line 153: Immunoblotting for NFATc1
What is NFATc1?
Line 127 the authors describe that “TRAP staining was performed after 7 days”, and line 187 “after 7 days TRAP-positive cells were counted”.
However, in the figure 1 the authors state that TRAP staining was performed after 5 days. Could explain?
Conclusions
Needs to be improved, to inform the readers to prevent and treatment of periodontitis.
Best regards
Author Response
- Title
S-PRG Eluate Suppresses Osteoclastogenesis through Downregulation of the MAPK Signaling Pathway.
The title needs to be changed. What is S-PRG?
Answer: We appreciate this comment. We changed the title to clarify “S-PRG” by sppeling it out as “Surface Pre-Reacted Glass-Ionomer”.
- Materials and Methods
Line 113: “…RAW264.7 OC precursor cells… What is this cell line?
Answer: RAW264.7 is a mouse macrophage cell line. We revised the manuscript to define the characteristics of the cell line (Line 116).
- Line 129: “cells…were counted under light microscopy…”
What is the microscopy used?
Answer: We added the information of the light microscope (EVOS™ XL Core Imaging System) (Line 139).
- Line 153: Immunoblotting for NFATc1
What is NFATc1?
Answer: We originally described NFATc1 in Discussion section (Line 349). However, we also added the information about NFATc1 in Materials and Methods section as well (Line 181).
- Line 127 the authors describe that “TRAP staining was performed after 7 days”, and line 187 “after 7 days TRAP-positive cells were counted”.
However, in the figure 1 the authors state that TRAP staining was performed after 5 days. Could explain?
Answer: Thank you for pointing out the typo errors. We carried out the TRAP staining at Day-5 for all experiments. We corrected the typo errors throughout the manuscript (Line 132 and 215).
- Conclusions
Needs to be improved, to inform the readers to prevent and treatment of periodontitis.
Answer: We are grateful for your comment that clearly improve the quality of manuscript. We revised the Conclusions to clarify that S-PRG eluate’s potential capacity to prevent and treat periodontal disease through regulating OC-genesis by synergic effects of multiple ions released from S-PRG (Line 414-421).
Reviewer 2 Report
Comments and Suggestions for Authors
This article described the role of S-PRG eluate solution on suppressing osteoclastogenesis and down-regulating the MAPK-related signals in vitro. TRAP staining, PCR, WB was applied to clearly detect the changes of osteoclastogenesis-related genes and proteins in RANKL-induced osteoclastogenesis precursor cells. There are some suggestions for the research as follows:
1. The authors mentioned the importance of the potential application of S-PRG in the tooth root caries, which is one of the novel points in the objectives rather than the treatment of periodontitis, although they have some common goals on the bone regeneration. However, the experiments didn’t involve in any characteristics of root caries (special bacteria, root caries animal model et al.), they only detect the changes of osteoclastogenesis in vitro. Nevertheless, previous study has demonstrated the decrease of OCs in periapical lesions when treated with S-PRG in mice. So, it seems the novelty is lower to average. Moreover, the point in conclusion is about the periodontitis, the authors should clearly demonstrate their ideas, and fulfill the ideas with adequate experiments.
2. In the background, the authors aim to elucidate the effect of multiple ions released from S-PRG filler on OC-genesis. However, they haven’t yet demonstrated the special function of these six ions.
3. The authors clearly showed the inhibition of OC-genesis when using S-PRG eluate solutions, the phosphorylation of MAPK proteins was also down-regulated at the same time. However, they haven’t yet proved the inhibition of OC-genesis was regulated via the MAPK signaling pathways without any inhibition experiments.
Comments on the Quality of English LanguageMinor editing of English language required
Author Response
- The authors mentioned the importance of the potential application of S-PRG in the tooth root caries, which is one of the novel points in the objectives rather than the treatment of periodontitis, although they have some common goals on the bone regeneration. However, the experiments didn’t involve in any characteristics of root caries (special bacteria, root caries animal model et al.), they only detect the changes of osteoclastogenesis in vitro. Nevertheless, previous study has demonstrated the decrease of OCs in periapical lesions when treated with S-PRG in mice. So, it seems the novelty is lower to average. Moreover, the point in conclusion is about the periodontitis, the authors should clearly demonstrate their ideas, and fulfill the ideas with adequate experiments.
Answer: Thank you for your intuitive comments. We agree with you. It is true that the indirect effect of S-PRG on OC-genesis in periapical lesions induced in rats (not mice) was reported previously (Reference 36). However, the latter study did only evaluate whether ion(s) released from S-PRG in the exposed pulp is(are) responsible for the inhibition of OC-genesis that occurs at a remote site of periapical bone. Therefore, we felt that only the in vitro OC-genesis assay can answer the above noted unsolved question, i.e., direct effect of S-PRG eluate on OC-genesis. Thus, the elucidation of direct effect of ions released from S-PRG on OC-genesis is novel report-worthy finding. We clearly demonstrated the direct impact of S-PRG eluate on OC-genesis by regulating MAPK signaling. Additionally, the other previously reported studies showed that S-PRG has capabilities to prevent and treat periodontal disease due to the anti-microbial or -inflammatory effects (Reference 24-26). Although microbiology is out of scope in our current study, our results indicated a direct impact of S-PRG on osteoclasts, and these findings draw a clear distinction from the previous reports. Thus, our findings are completely novel. To address these points, we revised the Discussion section of our manuscript (Line 397-408).
- In the background, the authors aim to elucidate the effect of multiple ions released from S-PRG filler on OC-genesis. However, they haven’t yet demonstrated the special function of these six ions.
Answer: We are appreciative of your relevant insightful comment. In response, we conducted the additional experiments to demonstrate the impact of individual ion among the 6 multiple ions released from S-PRG on OC-genesis. As depicted in Figure 8, we found anti-OC-genesis effects in all 5 ions (except Na+) and anti-resorption effect in 3 ions (Sr2+, F-, B3+). Thus we concluded that the synergic effects of the various ions present in S-PRG eluate account for the anti-OC-genesis effect by S-PRG eluate. We revised the manuscript accordingly to describe these findings (Line 294-303, Line 386-390).
- The authors clearly showed the inhibition of OC-genesis when using S-PRG eluate solutions, the phosphorylation of MAPK proteins was also down-regulated at the same time. However, they haven’t yet proved the inhibition of OC-genesis was regulated via the MAPK signaling pathways without any inhibition experiments.
Answer: Thank you for your comment. In response, the experiments using MAPK inhibitors were performed. Our additional results indicated that MAPK inhibitors (ERK, p38, and JNK) significantly suppressed OC-genesis compared to control (Figure 6). The manuscript was revised accordingly to include these results (Line 203-208, Line 274-284, Line 340-343).
Round 2
Reviewer 1 Report
Comments and Suggestions for Authors
Dear authors,
Congratulations on the improvement of the paper.
Author Response
We really appreciate your review of our manuscript.
Reviewer 2 Report
Comments and Suggestions for Authors
The authors have addressed most of the questions. However, as for the part to prove the inhibition of OC-genesis when using S-PRG eluate solutions was regulated via the MAPK signaling pathways. The newly added “3-5. MAPK inhibitors’ effects on OC-genesis” section could only prove the function of MAPK inhibitors on OC-genesis, which is not the main point in this manuscript. However, what the author should have demonstrated is the changes of S-PRG eluate solutions on OC-genesis when the MAPK pathways were inhibited or enhanced.
Comments on the Quality of English LanguageMinor editing of English language required
Author Response
Comment: The authors have addressed most of the questions. However, as for the part to prove the inhibition of OC-genesis when using S-PRG eluate solutions was regulated via the MAPK signaling pathways. The newly added “3-5. MAPK inhibitors’ effects on OC-genesis” section could only prove the function of MAPK inhibitors on OC-genesis, which is not the main point in this manuscript. However, what the author should have demonstrated is the changes of S-PRG eluate solutions on OC-genesis when the MAPK pathways were inhibited or enhanced.
Response: We are grateful for the comment. In response to reviewer 2's critique, we have revised the manuscript by performing additional experiment (new Fig 7 ). As shown in new Fig 7, we have demonstrated that S-PRG-mediated suppression of OC-genesis eluate may involve the modulation of the p38 and JNK activities, but not that of ERK.
Round 3
Reviewer 2 Report
Comments and Suggestions for Authors
The authors basically addressed the questions. They could enhance the description of the Discussion part.
Comments on the Quality of English LanguageMinor editing of English language required
Author Response
Comment: Minor editing of English language required.
Answer: Thank you for your additional review and comment. Our manuscript was revised with red-highlighted.